# Can adoption at an early age protect children at risk from depression in adulthood? A Swedish national cohort study

Anders Hjern,[1,2] Jesus Palacios,[3] Bo Vinnerljung[2,4]

[1]Centre for Health Equity Studies (CHESS), Karolinska Institutet, Stockholm University, Stockholm, Sweden
[2]Clinical Epidemiology/ Department of Medicine, Karolinska Institutet, Stockholm, Sweden
[3]Department of Developmental Psychology, University of Seville, Sevilla, Spain
[4]Department of Social Work, Stockholm University, Stockholm, Sweden

**Correspondence to**
Dr Anders Hjern; anders.hjern@ chess.su.se

## ABSTRACT

**Objective** Our aim was to investigate whether the risk of depression in adulthood in children raised by substitute parents from an early age differ by care arrangements.

**Methods** Register study in Swedish national cohorts born 1972–1981, with three study groups of children raised in adoptive or foster homes with care starting before the age of 2 years and a comparison majority population group. Cox regression estimated HRs of prescribed antidepressive medication and specialised psychiatric care with a diagnosis of depression in adulthood during 2006–2012.

**Results** Compared with the general population, long-term foster care carried the highest age-adjusted and sex-adjusted HR for both antidepressive medication, 2.07 (95% CI 1.87 to 2.28), and psychiatric care for depression, 2.85 (95% CI 2.42 to 3.35), in adulthood. Adults raised by adoptive parents were far more similar to the general population with HR of 1.19 (95% CI 1.00 to 1.43) for domestic and 1.13 (95% CI 1.08 to 1.18) for international adoption for antidepressive medication. Adjusting the analysis for school marks and income attenuated these risks more in the long-term foster care group.

**Conclusion** The study demonstrates the benefits of early adoption when substitute parents are provided for young children, and underlines the importance of improved educational support for children in foster care.

## INTRODUCTION

Early adverse experiences, particularly child abuse and neglect, have been linked to a number of negative outcomes, with longitudinal evidence showing the impact on physical[1] as well as on mental health.[2] Research on stress neurobiology is helping to gain a better understanding of the connections between early adversity (in the form of abuse and neglect, traumatic experiences and chronic exposure to toxic stress), its 'deep-seated' neurobiological alterations[3] and later mental health disturbances.

Literature reviews and meta-analytic evidence highlight the connection between early adversity and poor psychiatric outcomes, including depression. Between a quarter and

### What is already known on this topic?

► The risk of depression in adulthood is much increased in maltreated children.
► Foster care and adoption are the main types of care provided for children maltreated at an early age.
► Compared with those placed in foster care, adopted children have been shown to have better educational and employment outcomes in the adult Swedish population.

### What this study hopes to add?

► Adults with a history of long-term foster care had twofold rates of depression indicators compared with the general population.
► Adoption during infancy is associated with a risk of adult depression lower than long-term foster care.
► Poorer educational and employment outcomes may contribute to the higher risk of depression in former foster children.

a third of maltreated children meet criteria for major depression by their late 20s.[4] Compared with non-maltreated individuals, those with a history of abuse and neglect are twice as likely to develop both recurrent and persistent depressive episodes.[5]

In welfare societies, one common consequence of maltreatment in early childhood is placement into substitute families. Maintaining these children with their parents, or placing them in residential care, has been associated with a higher incidence of mental health problems, including anxiety, depression and conduct disorders.[6] At the same time, studies from many countries have consistently shown that growing up in substitute families does not eliminate the ill consequences of early adversity. Both adopted and fostered children exhibit more mental health problems—including depression—than those in the general population.[7] Swedish population

**BMJ**

studies have shown that, compared with majority population peers, young adults with a history of long-term foster care, as well as of domestic and intercountry adoption, present higher risk for various psychological and psychiatric disorders.[8–10] Considerable differences in educational achievement and labour market participation have also been reported.[11]

The higher levels of psychological and psychiatric morbidity in adulthood, as well as of poor educational performance indicators, have typically been assumed to be related primarily to exposure to familial risk factors before placement.[12] For the international adoptees, specific adverse consequences related to early neglectful institutional care have been reported.[13]

Age at placement has been demonstrated to be an important risk factor for negative outcomes later in life such as lower IQ, problem behaviour and emotional disturbances.[14 15] Type and quality of care also seem to play a role. Adoption provides more stability than foster care, with frequent changes of foster families and other out-of-home care arrangements being inherent to long-term foster care.[16 17] This instability causes fostered children to develop behavioural and mental health problems. Foster care instability is also present in Sweden, the setting of this study. Even for long-term placements that last from early years until the beginning of adolescence, every fourth breaks down during the first teenage years.[18] Adopted children in Sweden may also experience placement instability at some time before age of majority, but the prevalence (3%) is minuscule compared with long-term foster care.[19] In their analysis of the impact of placement stability on the well-being of foster children, Rubin et al[20] concluded that foster children experience placement instability unrelated to their baseline problems, and that this instability has per se a significant impact on their behaviour and well-being.

In Sweden, domestic adoption was common in the mid-1900s, including around 1% of birth cohorts, often children born out of wedlock by young mothers. During the 1960s, new welfare policies that provided economic support for single mothers and a more liberal abortion policy gradually decreased the number of children available for adoption. Since the mid-1970s, domestic adoptions have been rare in Sweden and foster care has become the main form of long-term substitute care for young children who cannot be reunited with their birth parents.[21 22] International adoption in Sweden was initiated in the 1960s and peaked in the 1970s when birth cohorts in Sweden included 1.5%–2.0% internationally adopted children, at that time the highest per capita proportion in the world.[8]

The aim of this study was to exploit this period of drastic changes in Swedish adoption practices to investigate whether the risk of depression in adulthood differs between different forms of substitute care for children placed in care in infancy, and whether risk patterns differ between the lighter forms of depression, treated outside of hospitals (ie, antidepressive medication) and more severe forms, treated in hospital care (ie, specialised psychiatric care with a diagnosis of depression). Taking into account the well documented differences in school performance and labour market participation between adopted and long-term foster children in Sweden,[11] we also wanted to explore if these factors influence their risk of depression.

## METHODS

This study is based on information from Swedish national registers, containing data with high validity and low attrition rates.[23 24] These registers are based on the unique personal identity number assigned at birth (or time of immigration) to all Swedish residents. Data from different registers can be linked using these identity numbers.

The study population comprises individuals born 1972–1981 who, according to the Register of the Total Population (RTP), were alive and resident in Sweden on 31 December 2005. This population was followed up from 1 January 2006 to 31 December 2012, at age 25–41 years. Biological and/or adoptive parents of these individuals were identified in the multigeneration register. Information about region of birth, year of adoption, sex and year of birth in RTP was linked to the study subjects and their parents. Data on age at first placement in out-of-home care (OHC) and time spent in OHC before age 18 years was retrieved from the National Child Welfare Register.

Based on this information, we created three study groups of children who had entered OHC or been adopted before the age of 2 years: *domestic adoptees* (n=618) were Swedish-born and had two registered Swedish-born adoptive parents. *International adoptees* (n=8878) fulfilled the criteria of being born outside of western Europe, having at least one Swedish-born adoptive parent but no birth parent in the multigeneration register, and having entered Sweden before their second birthday; *long-term foster care* subjects (n=1 115) had no record of adoption and a total time in OHC of at least 10 years (median of 15.2 years) in care before age 18 years. A comparison population labelled *majority population* (n=930 944), all Swedish-born individuals with at least one Swedish-born birth parent and no adoptive parents in the multigeneration register, and with no record of placement in OHC before the age of 18 years.

### Depression

Two indicators of depression were created for the follow-up 2006–2012[1]: first indication of dispensed antidepressive medication (AC-code N06A) was obtained from the Swedish Prescribed Drug Register,[2] first entry in outpatient specialised care or a hospital discharge with a diagnosis of depressive disorder (F32-F39 in 10th revision of the International Classification of Disease) into the National Patient Discharge Register.

## Sociodemographic end educational covariates

Data on age, gender, disposable income, domicile and indicators of labour market participation were retrieved from the 2005 Longitudinal Integration Database for Health Insurance and Labour Market Studies. *Disposable income* was divided into quintiles, and included all registered sources of income deducted by taxes, and thereafter divided by consumer units in the household according to a formula developed by Statistics Sweden. *Domicile* was split into three categories and defined by the place of residence in 2008: 'city' referred to the metropolitan areas of Sweden's three largest cities, Stockholm, Gothenburg and Malmo, 'town' covered other predominately urban communities, and 'rural' covered the remainder. *Labour market participation* was defined as having an income from employment/self-employment in November 2005 or having received societal benefits as an active student that year. *School marks* in grade 9 (final year of compulsory school, 15–16 years) were retrieved from the National School Register, which is administered jointly by the Swedish School Authority and Statistics Sweden. The grading system during 1987–1996 nationally followed a Gaussian distribution and consisted of a scale from 1 (poor) to 5 (excellent).

## Statistical analyses

Cox proportional hazards models were used to estimate HRs and corresponding 95% CI for dispense of antidepressive medication and hospital specialised care with a diagnosis of depression as defined above. Person time of follow-up was accumulated from January 2006 until the date of the first dispense/date of first admission/visit, date of death from the National Cause of Death Register or end of follow-up in December 2013.

**Table 1** Characteristics of the study population

| | Domestic adoptees | International adoptees | Foster care | General population |
|---|---|---|---|---|
| | n=618 | n=8778 | n=1115 | 930944 |
| **Sex** | | | | |
| Men | 56.1 | 38.5 | 49.4 | 51.4 |
| Women | 43.9 | 61.5 | 50.6 | 48.6 |
| Age at adoption/entry into foster care | | | | |
| 0–12 months | 90.3 | 42.5 | 45.5 | – |
| 13–24 months | 9.7 | 57.5 | 54.5 | – |
| Year of birth | | | | |
| 1972–1974 | 50.3 | 23.1 | 30.2 | 32.6 |
| 1975–1977 | 27.7 | 32.8 | 26.0 | 29.4 |
| 1978–1981 | 22.0 | 44.1 | 43.8 | 38.0 |
| Marks in ninth grade | | | | |
| Mean average | 2.99 | 3.08 | 2.55 | 3.21 |
| Education in 2005 | | | | |
| ≤9 years | 10.0 | 10.2 | 35.2 | 9.3 |
| 10–12 years | 55.2 | 50.6 | 54.9 | 49.1 |
| 13+ years | 34.8 | 39.2 | 9.9 | 41.5 |
| Income in quintiles in 2005 | | | | |
| First | 11.8 | 14.6 | 26.1 | 13.2 |
| Second | 17.3 | 17.8 | 24.2 | 18.3 |
| Third | 19.4 | 20.5 | 23.1 | 20.2 |
| Fourth | 23.6 | 24.4 | 16.2 | 22.7 |
| Fifth | 27.8 | 22.6 | 10.5 | 25.6 |
| Employed November 2005 | 79.0 | 69.7 | 53.3 | 73.4 |
| Student in 2005 | 13.0 | 24.8 | 12.3 | 20.1 |
| Geographic residency | | | | |
| City | 42.9 | 52.8 | 37.7 | 42.9 |
| Town | 43.9 | 39.1 | 44.6 | 43.9 |
| Rural | 13.3 | 8.1 | 8.5 | 13.3 |

**Table 2** Depression indicators by sociodemographic variables (percentages)

| | Medication | | Hospital care | |
|---|---|---|---|---|
| | **Men** | **Women** | **Men** | **Women** |
| **Study groups** | | | | |
| General population | 12.3 | 21.6 | 3.4 | 5.8 |
| Domestic adoptees | 12.1 | 29.2 | 3.2 | 5.2 |
| International adoptees | 14.1 | 24.0 | 5.5 | 8.1 |
| Foster care | 26.8 | 39.1 | 11.8 | 13.5 |
| **Year of birth** | | | | |
| 1972–1974 | 12.2 | 22.5 | 3.1 | 5.3 |
| 1975–1977 | 12.4 | 21.7 | 3.4 | 5.7 |
| 1978–1981 | 12.4 | 21.0 | 3.7 | 6.3 |
| **Highest education in 2005** | | | | |
| ≤9 years | 19.0 | 29.4 | 7.6 | 12.8 |
| 10–12 years | 12.2 | 23.1 | 3.4 | 6.2 |
| 13+ years | 9.1 | 16.8 | 2.1 | 4.3 |
| **Student and/or employed in 2005** | | | | |
| No | 24.7 | 35.2 | 9.3 | 12.7 |
| **Income in quintiles in 2005** | | | | |
| First | 17.1 | 22.5 | 5.9 | 8.7 |
| Second | 15.1 | 25.1 | 4.5 | 7.2 |
| Third | 13.9 | 22.3 | 4.0 | 5.9 |
| Fourth | 11.7 | 18.5 | 3.1 | 4.6 |
| Fifth | 8.8 | 14.7 | 1.9 | 3.5 |
| **Geographic residency** | | | | |
| City | 12.2 | 21.1 | 3.4 | 5.8 |
| Town | 12.4 | 22.2 | 3.4 | 5.9 |
| Rural | 12.3 | 22.4 | 3.4 | 5.7 |
| Total | 12.3 | 21.7 | 3.4 | 5.8 |

The modelling of early childhood determinants of adult mental health is complex, since trajectories potentially involve intermediate factors that are products of the early childhood environment and determinants of adult mental health. In a previous study,[11] we have shown that the educational achievement and income of young adult men raised in foster care in Sweden is considerably lower than that of men in the general population with the same test scores on IQ tests, suggesting that educational achievement and income are important components in trajectories from foster care to adulthood. In this study, we therefore considered model 1, adjusted only for gender, and urban/rural residency as category variables and year of birth as a continuous variable, as the main analysis of the effects of care in early childhood. To study the effect of care, after adjustment for potential consequences of type of care, we also created a model 2 adjusted for grade point averages at age 15–16 years and a model 3 further adjusted for employment/

income in adult age. All analyses were conducted using SPSS V.24.0.

## RESULTS

Table 1 shows the characteristics of the study groups. International adoptees were more often females in comparison with the other study groups. The domestic adoptees were the youngest when entering care (90% during their first year of life), with international adoptees more often being adopted between their first and second birthday (46%). As expected from the policy change described above, the domestic adoptees were more often than the other study groups born in 1972–1974 and the foster children more often during 1975–1981. The domestic adoptees had the highest income and employment rates of the study groups. Foster children had the lowest school marks in grade 9, the lowest educational achievement and the lowest income. Domestic and international adoptees had mean grade point averages and

**Table 3** Cox regression models of indicators of depression and different forms of substitute care

|  | Model 1 | Model 2 | Model 3 |
|---|---|---|---|
|  | HR (95% CI) | HR (95% CI) | HR (95% CI) |
| Antidepressive medication |  |  |  |
| General population | 1 | 1 | 1 |
| Domestic adoptees | 1.19 (1.00 to 1.43) | 1.12 (0.93 to 1.34) | 1.13 (0.94 to 1.35) |
| International adoptees | 1.13 (1.08 to 1.18) | 1.06 (1.01 to 1.11) | 1.05 (1.00 to 1.10) |
| Foster children | 2.07 (1.87 to 2.28) | 1.64 (1.47 to 1.83) | 1.44 (1.29 to 1.61) |
| Hospital care |  |  |  |
| General population | 1 | 1 | 1 |
| Domestic adoptees | 0.93 (0.63 to 1.38) | 0.85 (0.57 to 1.27) | 0.86 (0.58 to 1.29) |
| International adoptees | 1.45 (1.34 to 1.57) | 1.32 (1.22 to 1.44) | 1.30 (1.19 to 1.41) |
| Former foster children | 2.85 (2.42 to 3.35) | 2.01 (1.68 to 2.41) | 1.62 (1.35 to 1.94) |

Model 1 is adjusted for gender, year of birth and urban/rural residency.
Model 2 is also adjusted for mean grade point averages in ninth grade.
Model 3 is also adjusted for income in quintiles and employment/being a student in young adulthood.

educational achievement in between the comparison population and the foster children.

As table 2 demonstrates, the accumulated incidence of antidepressive medication was 12% in men and 21% in women in the general population. The highest incidences were found among the former foster children, with 27% for men and 39% for women. For hospital psychiatric care with a diagnosis of depression, the accumulated incidence was 3% in men and 6% in women in the general population. Former foster children had the highest incidences also for this outcome; 13% for women and 12% for men.

Cox regression models of the effects of type of care on depression outcomes are presented in table 3. In the main analysis, presented in model 1, we adjusted for year of birth, gender and rural/urban residency only. Compared with the general population, individuals in the foster care group had the highest HRs for both outcomes, 2.07 (95% CI 1.87 to 2.28) for antidepressive medication, and 2.85 (95% CI 2.42 to 3.35) for hospital psychiatric care with a diagnosis of depression. International and domestic adoptees had similar risks for antidepressive medication, HRs 1.13 (95% CI 1.08 to 1.18) and 1.19 (955 CI 1.00 to 1.43), respectively, while the HR for psychiatric care for depression was higher in international adoptees, 1.45 (95% CI 1.34 to 1.57) compared with 0.93 (95% CI 0.63 to 1.38) in the domestic adoptees. Interaction analyses could not detect any significant gender differences for any of these two outcomes.

Adjusting the analysis for factors on the trajectory from childhood to adulthood, school marks in model 2 and income and employment in model 3, attenuated these risks more in the foster care group, suggesting that some of the excess risks for depression in the former foster children was indeed associated with these intermediate factors. HRs, however, remained more elevated for former foster children than for the adoption groups and the general population.

## DISCUSSION

This national cohort study of adults raised by adoptive or foster parents demonstrates that, compared with foster care, adoption during the infancy years is associated with a lower risk of antidepressive medication and psychiatric care with a diagnosis of depression in adulthood. Education, employment and income, as possible consequences of type of care, attenuated these risks more in the foster care group.

Previous research on Swedish adults raised in substitute families have shown that foster children consistently have a higher consumption of psychiatric care and psychotropic drugs, and a higher risk of suicidality than adoptees.[25–27] The pattern demonstrated in this study suggests that a higher incidence of depressive disorders may be an important factor behind the excess risk of suicide previously found in former foster children compared with international adoptees in Sweden.[25]

In the foster care group, the HRs were strongly attenuated by school marks and income levels, although the percentage of those with higher academic qualifications and better income is significantly lower for the foster care group. Previous Swedish studies have shown that, despite a similar cognitive competence,[9] adoptees perform better in school, reach higher educational levels and have higher labour market participation than adults raised in foster care, with the international adoptees doing particularly well.[11] Thus, family related environmental differences in cognitive and educational support before and during the school years, and the consequences on labour market participation of this 'stunted' educational achievement for the former foster children may be one mechanism that can explain their higher rate of depression.[28]

The risk estimates for the foster children and the international adoptees were particularly high for the more severe outcome of hospital psychiatric care, compared with antidepressive medication. This is in line with

previous studies, where depression associated with childhood adversity has been linked to a greater risk of recurrent and persistent depressive episodes, and also to less benefits from treatment.[5]

The results of this study suggests that early adoption can limit the risk of depression associated with early childhood adversity. Lacking detailed individual data, we hypothesise that an important mechanism is that adoption provides better stability and a more predictable childhood home base—both for children and substitute parents.[29] From a developmental perspective, safety, security, stability and nurturance in out-of-home placements are key requisites for the recovery from past adversities and the well-being of looked after children.[30] The importance of stability is highlighted by studies showing that an increase in the number of OHC placement experiences predicts a greater rate of mental health difficulties.[31 32] The well-known difficulties for youth transitioning from OHC to independent living also seem to include substantial problems with access to adult mental health services.[33]

Our findings can be putatively linked to neurobiological studies showing the connection between early adversity and the HPA system, with persistent high levels of cortisol having been linked to adult depression in animal models.[34] Laurent et al[35] have demonstrated that a high-quality caring environment has the potential to normalise and stabilise the daily pattern of cortisol levels in early childhood.

### Strengths and limitations

The main strength of this study is the use of the Swedish national registers that allows for long-term follow-up with minimal attrition and high-quality data. The main limitation is the lack of information about familial risk factors, that is, hereditary, fetal and early childhood exposures in the study groups and the reason for why the children were taken into foster care or were adopted. In a previous study, where we had access to more information about the birth parents, we showed that the birth parents of national adoptees as well as children in long-term foster care have much higher cumulated incidences of psychiatric disorders and substance abuse than the general population, with the long-term foster children having the highest incidences. Thus, it seems probable that fetal exposures to substances and genetic risk factors for depression explain some of the differences between the study groups.[11] The use of a unique window in Swedish child welfare history, however, when domestic adoptions were replaced with foster care as the default choice for placements in maltreated young children can be expected to limit the influence of such confounding.

Another type of potential confounding would have to have been considered if adopted children and/or foster children were offered targeted psychiatric services that would increase the chance of receiving psychiatric care and having been being prescribed antidepressive medication in adulthood. Such targeted services, however, are not in place in Sweden today and were not in place when

the studied populations were children. This and other studies indicate that such services should be considered.[36]

### CONCLUSIONS

This study suggests that early adoption can protect children from some or even all of the risk of depression associated with early childhood adversity. Further studies within the foster care group are needed to elucidate the protective effect of placement stability.

**Contributors** AH came up with the idea of this manuscript, made the register linkages, created the data set for the analyses, made all statistical analyses and wrote the first draft of the manuscript. JP and BV both contributed to the theoretical framework of the study, the interpretation of the results and the writing of the manuscript. All three authors have read and approved the final manuscript.

**Funding** This study was supported by a grant from the Bank of Sweden Tercentenary Foundation P10-0514:1.

**Competing interests** None declared.

**Patient consent for publication** Not required.

**Ethics approval** This study was approved by the ethics committee in the Stockholm region (No. 2014/415-31/5).

**Provenance and peer review** Not commissioned; externally peer reviewed.

**Data sharing statement** The register data used in this study is protected by Swedish legislation and cannot be shared by researchers not involved in the study.

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
