## [Reviewer comments · BMJ Paediatrics Open]

ARTICLE DETAILS

TITLE (PROVISIONAL)	Can adoption at an early age protect children at risk from depression in adulthood? A Swedish National Cohort Study.
AUTHORS	Hjern, Anders; Palacios, Jesus; Vinnerljung, Bo

VERSION 1 – REVIEW

REVIEWER	Reviewer name: Peter Flom Institution and Country: Peter Flom Consulting, USA Competing interests: None
REVIEW RETURNED	02-Aug-2018

GENERAL COMMENTS	I confine my remarks to statistical aspects of this paper. They were generally fine, but I have some issues to resolve before I can recommend publication. One concern is whether the people in the different groups were different before they were placed. This is more of a concern for comparing with the general population, since the difference in adoption and foster care seems to be due to political considerations. Still, as long as some kids were adopted and some in foster care, it could exist there too. The authors do note it in limitations. If there is data on various characteristics of the people, then perhaps some form of matching or propensity scores could be used. I also think model 3 should be the focal point, rather than model 1. p. 5 line 19 - be careful about causal language. p. 9 line 54 - do not categorize continuous variables. This lowers power and increases type 1 errors.
---

REVIEWER	Reviewer name: Shanti Raman Institution and Country: South Western Sydney Local Health District & University of New South Wales, Australia Competing interests: I declare I have no competing interests
REVIEW RETURNED	09-Oct-2018

GENERAL COMMENTS	This is a really important topic and a very timely exploration of the issue. The findings of this large study have widespread implications for public health and social policy in the western world. The data and analysis you have access to in Sweden with your Swedish National Registers is enviable. There are a couple of minor grammatical and punctuation edits required, such as a few unnecessary apostrophes eg “1900’s”, but that is a minor quibble and easily fixed with sharp editing. Edits
---

	P6: 2nd para: Taking into account the well documented differences in school performance and labor market participation between adopted and longterm foster children in Sweden, change to: we also wanted to explore if these factors influence the risk of depression. P8: last para: ttrajectories (remove extra "t")
--	--

VERSION 1 – AUTHOR RESPONSE

Reviewer: 1

Comments to the Author

I confine my remarks to statistical aspects of this paper. They were generally fine, but I have some issues to resolve before I can recommend publication.

1. One concern is whether the people in the different groups were different before they were placed. This is more of a concern for comparing with the general population, since the difference in adoption and foster care seems to be due to political considerations. Still, as long as some kids were adopted and some in foster care, it could exist there too. The authors do note it in limitations. If there is data on various characteristics of the people, then perhaps some form of matching or propensity scores could be used.

Comment: Already in our original manuscript we acknowledged that the main problem with our study is the lack of information about the reasons for why children were adopted or taken into foster care and information about their biological parents. In this revised manuscript we have developed this a little more with information from a previous study where we had more information about parental characteristics. See above for new text,

2. I also think model 3 should be the focal point, rather than model 1.

Usually, the last model in a series of models like those in our analysis, is considered the fully adjusted model and reported as the main results. As we explain in the methods section, however, we have previously shown that children in long term foster care do not reach their educational potential as defined by their IQ, while adoptees do. Thus, educational achievement and the vocational career that follows, must be considered to be on the pathway of what we are studying here, rather than being considered confounders. Thus we still think that Model 1 is the best focal point of the reporting from this study and that Models 2 and 3 are best treated as explanatory models.

p. 5 line 19 - be careful about causal language.

Comment: We are quoting Rubin et al here, this is not our own conclusion. Rephrasing this in a less causal manner would make the citation incorrect.

p. 9 line 54 - do not categorize continuous variables. This lowers power and increases type 1 errors.

Comment: There is no line 54 on page 9 in the manuscript, but I imagine you are thinking about the coding of year of birth in the statistical analysis. Although we presented year of birth as a three category variable in the descriptive table, to save space, it was included as a continuous variable in the statistical models. We have now clarified this in the text about the statistical analysis.

Reviewer: 2

Comments to the Author

General Comments

This is a really important topic and a very timely exploration of the issue. The findings of this large study have widespread implications for public health and social policy in the western world. The data and analysis you have access to in Sweden with your Swedish National Registers is enviable. There are a couple of minor grammatical and punctuation edits required, such as a few unnecessary apostrophes eg "1900's", but that is a minor quibble and easily fixed with sharp editing.

Comment ; Thank you Shanti!

Edits

P6: 2nd para: Taking into account the well documented differences in school performance and labor market participation between adopted and longterm foster children in Sweden, change to: we also wanted to explore if these factors influence the risk of depression.

Comment:

Changed to: we also wanted to explore if these factors influence their risk of depression

P8: last para: ttrajectories (remove extra "t")

Comment: Sorry, that was sloppy! Made the change....

The manuscript is not being considered for publication elsewhere, or has previously been published or accepted for publication.